

# Feature selection by integrating document frequency with genetic algorithm for Amharic news document classification

Demeke Endalie[1], Getamesay Haile[1] and Wondmagegn Taye Abebe[2]

[1] Faculty of Computing and Informatics, Jimma Institute of Technology, Jimma, Oromia, Ethiopia
[2] Faculty of Civil and Environmental Engineering, Jimma Institute of Technology, Jimma, Oromia, Ethiopia

## ABSTRACT

Text classification is the process of categorizing documents based on their content into a predefined set of categories. Text classification algorithms typically represent documents as collections of words and it deals with a large number of features. The selection of appropriate features becomes important when the initial feature set is quite large. In this paper, we present a hybrid of document frequency (DF) and genetic algorithm (GA)-based feature selection method for Amharic text classification. We evaluate this feature selection method on Amharic news documents obtained from the Ethiopian News Agency (ENA). The number of categories used in this study is 13. Our experimental results showed that the proposed feature selection method outperformed other feature selection methods utilized for Amharic news document classification. Combining the proposed feature selection method with Extra Tree Classifier (ETC) improves classification accuracy. It improves classification accuracy up to 1% higher than the hybrid of DF, information gain (IG), chi-square (CHI), and principal component analysis (PCA), 2.47% greater than GA and 3.86% greater than a hybrid of DF, IG, and CHI.

## INTRODUCTION

Amharic is an Ethiopian language that belongs to the Semitic branch of the Afro-Asian language family. Amharic is the official working language of the Ethiopian Federal Democratic Republic, and it is the world's second most spoken Semitic language after Arabic, with approximately 22 million speakers according to *Hagos & Mebrahtu (2020)* and *Wakuma Olbasa (2018)*. Amharic is classified as a low-resource language when compared to other languages such as English, Arabic, and Chinese (*Gereme et al., 2021*). Due to this, a significant amount of work is required to develop many Natural Language Processing (NLP) tasks to process this language.

Text processing has become difficult in recent years due to the massive volume of digital data. The curse of dimensionality is one of the most difficult challenges in text processing (*Aremu, Hyland-Wood & McAree, 2020*). Feature selection is one of the techniques for dealing with the challenges that come with a large number of features text classification is a natural language processing task that requires text processing. Text classification

Corresponding author
Demeke Endalie,
demeke.endalie@ju.edu.et

performance is measured in terms of classification accuracy and the number of features used. As a result, feature selection is a crucial task in text classification using machine learning algorithms.

Feature selection aims at identifying a subset of features for building a robust learning model. A small number of terms among millions shows a strong correlation with the targeted news category. Works in *Tuv et al. (2009)* address the problem of defining the appropriate number of features to be selected. The choice of the best set of features is a key factor for successful and effective text classification (*Hartmann et al., 2019*). In general, redundant and irrelevant features cannot improve the performance of the learning model rather they lead to additional mistakes in the learning process of the model.

Several feature selection methods were discussed to improve Amharic text classification performance (*Endalie & Tegegne, 2021*; *Kelemework, 2013*). Existing feature selection methods for Amharic text classifications employ filter approaches. The filter approach select features based on a specific relevance score. It does not check the impact of the selected feature on the performance of the classifier. Additionally, the filter feature selection technique necessitates the setting up of threshold values. It is extremely difficult to determine the threshold point for the scoring metrics used to select relevant features for the classifier (*Salwén, 2019*; *Akhter et al., 2022*). A better feature method based on classifier performance improves classification accuracy while decreasing the number of features.

As a result, this study presents a hybrid feature selection method that combines document frequency with a genetic algorithm to improve Amharic news text classification. The method can also help us to minimize the number of features required to represent each news document in the dataset. The proposed feature selection method selects the best possible feature subset by considering individual feature scoring and classifier accuracy. The contributions of this study are summarized as follows.

1. Propose a feature selection method that incorporates document frequency and a genetic algorithm.
2. Prove that the proposed feature selection method reduces the number of representative features and improves the classification accuracy over Amharic new document classification.

The rest of the paper is organized as follows: "Related Works" is the description of the literature review. "Materials and Methodology" describes the feature selection technique and methodology used in this work, which is based on document frequency and genetic algorithms. "Experiment" presents and discusses the experimental results. "Conclusion" focuses on the conclusion and future work.

## RELATED WORKS

The accuracy of classifier algorithms used in Amharic news document classification is affected by the feature selection method. Different research has attempted to overcome the curse of dimensionality by employing various feature selection techniques. The following

are some of the related feature selection works on Amharic and other languages document classification.

*Endalie & Tegegne (2021)* proposed a new dimension reduction method for improving the performance of Amharic news document classification. Their model consists of three filter feature selection methods *i.e.*, IG, CHI, and DF, and one feature extractor *i.e.*, PCA. Since a different subset of features is selected with the individual filter feature selection method, the authors used both union and intersection to merge the feature subsets. Their experimental result shows that the proposed feature selection method improves the performance of Amharic news classification. Even though the weakness of one feature selection method is filled by the strength of the other, the feature selection method used in their model does not consider the interaction among features on the classifier performance.

*Endalie & Haile (2021)* proposed a hybrid feature selection method for Amharic news text classification by integrating three different filter feature selection methods. Their feature selection method consists of information gain, chi-square, and document frequency. The proposed feature selection improves the performance of Amharic news text classification by 3.96%, 11.16%, and 7.3% more than that of information gain, chi-square, and document frequency, respectively. However, the dependency among terms (features) is not considered in their feature selection method.

Feature selection algorithms were proposed in *Mera-Gaona et al. (2021)* to analyze highly dimensional datasets and determine their subsets. Ensemble feature selection algorithms have become an alternative with functionalities to support the assembly of feature selection algorithms. The performance of the framework was demonstrated in several experiments. It discovers relevant features either by single FS algorithms or by ensemble feature selection methods. Their experimental result shows that the ensemble feature selection performed well over the three datasets used in their experiment.

*Ahmad et al. (2020)* proposed a more accurate ensemble classification model for detecting fake news. Their proposed model extracts important features from fake news datasets and then classifies them using an ensemble model composed of three popular machine learning classifiers: Decision Tree, Random Forest, ETC. Ensemble classifiers, on the other hand, require an inordinate amount of time for training.

*Marie-Saintea & Alalyani (2020)* proposed a new bio-inspired firefly algorithm-based feature selection method for dealing with Arabic speaker recognition systems. Firefly algorithm is one of the wrapper approaches to solving nonlinear optimization problems. They proved that this method is effective in improving recognition performance while reducing system complexity.

In *Muštra, Grgić & Delač (2012)*, the authors explore the use of wrapper feature selection methods in mammographic images for breast density classification. They used two mammographic image datasets, five wrapper feature selection methods were tested in conjunction with three different classifiers. Best-first search with forwarding selection and best-first search with backward selection was the most effective methods. These feature selection methods improve the overall performance by 3% to 12% across different classifiers and datasets.

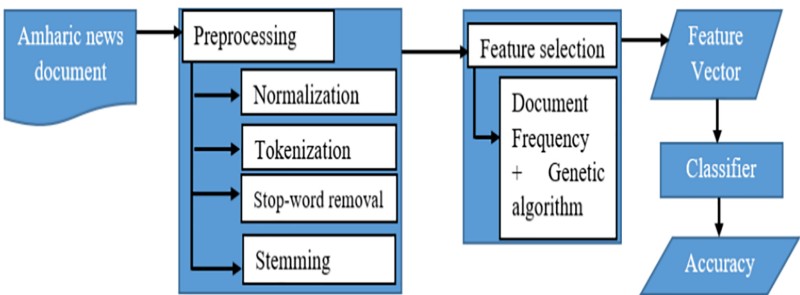

**Figure 1 The architecture of the proposed Amharic text classifier.**

The work of *Zhu et al. (2019)* proposed a novel unsupervised feature selection technique, feature selection-based feature clustering that uses similarity-based feature clustering (FSFC). The results of feature clustering based on feature similarity are used by FSFC to reduce redundant features. First, it groups the characteristics based on their similarity. Second, it chooses a representative feature from each cluster that includes the most interesting information among the cluster's characteristics. Their experimental results reveal that FSFC not only reduces the feature space in less time but also improves the clustering performance of K-means considerably.

# MATERIALS AND METHODOLOGY

The collection of Amharic news documents served as the starting point for the suggested feature selection technique for Amharic news document classification. The proposed document classifier consists of preprocessing, document representation, feature selection, and a classifier module. Figure 1 depicts the overall architecture of the proposed classifier in this study.

## Data preprocessing

Real-world data generally contains noises, missing values, and maybe in an unusable format that cannot be directly used for machine learning models. Data preprocessing is required for cleaning the data and making it suitable for a machine learning model and also helps us to increase the accuracy and efficiency of a machine learning model (*Iliou, Anagnostopoulos & Nerantzaki, 2015*). Amharic is one of the languages which is characterized by complex morphology (*Gasser, 2011*). Separate data pre-processing is required because Amharic has its own set of syntactical, structural, and grammatical rules. To prepare the raw Amharic news document for the classifier, we performed the following preprocessing tasks.

### Normalization

There are characters in Amharic that have the same sound but no clear-cut rule in their meaning difference. The number of features increases when we write the name of an object or concept with different alphabets/characters. We create a list of characters and their corresponding canonical form used by this study. For instance, characters such as ሀ፣ሃ፣ኀ፣ኻ፣ሐ፣ሓ, ሰ፣ሠ, እ፣ኣ፣ዐ፣ዒ, ጸ፣ፀ have the same sound and meaning. Table 1 shows the

**Table 1 List of consonants normalized in the study.**

| Canonical form | Characters to be replaced |
|---|---|
| hā(ሀ) | hā(ኀ፣ኃ፣ኈ፤ሐ፤ሓ) |
| se(ሰ) | se(ሠ) |
| ā(አ) | ā(ኣ፤ዐ፤ዓ) |
| ts'e(ጸ) | ts'e(ፀ) |
| wu(ው) | wu(ዉ) |
| go(ጐ) | go(ጎ) |

normalization of Amharic characters having the same sound with different symbolic representations (*Endalie, Haile & Gastaldo, 2021*).

### Stop-word removal

Words in the document do not have equal weight in the classification process. Some are used to fill the grammatical structure of a sentence or do not refer to any object or concept. Common words in English text like, a, an, the, who, be, and other common words that bring less weight are known as stop-words (*Raulji & Saini, 2016*). We used the stop-word lists prepared by *Endalie & Haile (2021)*. We remove those terms from the dataset before proceeding to the next text classification stage.

### Stemming

Stemming is the process of reducing inflected words to their stem, base, or root form. Amharic is one of the morphological-rich Semitic languages (*Tsarfaty et al., 2013*). Due to this, different terms can exist with the same stem. Stemming helps us to reduce morphological variant words to their root and reduce the dimension of the feature space for processing. For example, ቤት "House" (ቤቱ, ቤቶች, ቤታችን, ቤቶቻችን, ቤታቸው, ቤቶቻቸው) into their stem word ቤት. In this paper, we used Gasser's HornMorpho stemmer (*Gasser, 2011*). HornMorpho is a Python program that analyzes Amharic, Oromo, and Tigrinya words into their constituent morphemes (meaningful parts) and generates words, given a root or stem and a representation of the word's grammatical structure. It is rule-based that could be implemented as finite-state transducers (FST). We adopt this stemmer because it has 95% accuracy and is better as compared with other stemmers (*Gebreselassie et al., 2018*).

## Document representation

To transform documents into feature vectors we used Bag-of-Word (BOW) method. The BOW is denoted with vector space model (VSM). In this type of document representation, documents are represented as a vector in n-dimensional space, where n is the number of unique terms selected as informative from the corpus (*Miao & Niu, 2016*). The weight of each term can be calculated by one of the term weighting schemes like Boolean value, term frequency, inverse document frequency, or term frequency by inverse document frequency. In the VSM document, $D_i$ can be represented as $[W_{i1}, W_{i2} \ldots W_{ij} \ldots W_{in}]$ where $W_{ij}$ is the weight computed by one of the above weighting scheme's values

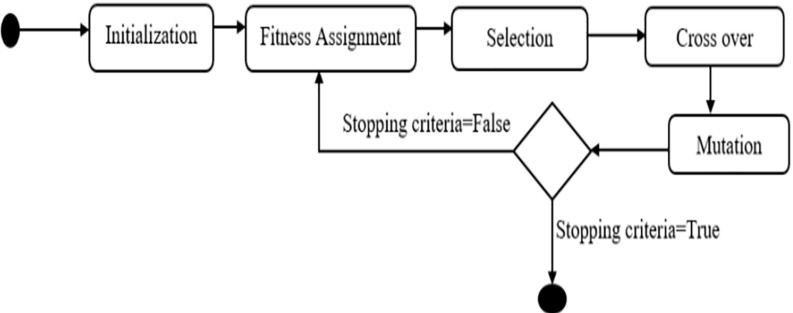

**Figure 2 State chart for feature selection using genetic algorithm.**

of the jth term in the n-dimensional vector space. Despite the disadvantage of BOW stated above it is still the dominant document representation technique used for document categorization in literature (*Bharti & Singh, 2015*; *Said, 2007*) due to its simplicity, best feature representation, and efficiency.

## Feature selection

Feature selection is the process of choosing a small subset of relevant features from the original features by removing irrelevant, redundant, or noisy features. Feature selection is very important in pattern recognition and classification. Feature selection usually leads to better learning accuracy, lower computational cost, and model interpretability. In this section, we define the document frequency, genetic algorithm, and proposed hybrid feature selection for Amharic text classification.

## Document frequency

Document frequency (DF) counts the number of documents which contains the given term. DF is determined as words scoring. DF value greater than a threshold are used for text classification. The fundamental idea behind DF is that terms that are irrelevant to the classification are found in fewer documents. DF is determined as *Hakim et al. (2014)*.

$$DF(t_i) = \sum_{i=1}^{m}(Ai) \tag{1}$$

where m is the number of documents and Ai is the occurrence of a term in document i.

## Genetic algorithm

The genetic algorithm is one of the most advanced feature selection algorithms (*Wang et al., 2021*). It is a stochastic function optimization method based on natural genetics and biological evolution mechanics. Genes in organisms tend to evolve over generations to better adapt to their surroundings. Figure 2 illustrates a statechart diagram of feature selection using a genetic algorithm.

A genetic algorithm consists of operators such as initialization, fitness assignment, selection, crossover, and mutation. Following that, we go over each of the genetic algorithm's operators and parameters.

### Initialization operator

The first step is to create and initialize individuals in the population. Individuals' genes are randomly initialized because a genetic algorithm is a stochastic optimization method.

### Fitness assignment operator

Following the initialization, we must assign a fitness value to each individual in the population. We train each neural network with training data and then evaluate its performance with testing data. A significant selection error indicates poor fitness. Individuals with higher fitness are more likely to be chosen for recombination. In this study, we used a rank-based fitness assignment technique to assign fitness values to each individual (*Zaman, Paul & Azeem, 2012*).

### Selection operator

Following the completion of a fitness assignment, a selection operator is used to select individuals to be used in the recombination for the next generation. Individuals with high fitness levels can survive in the environment. We used the stochastic sampling replacement technique to select individuals based on their fitness, where fitness is determined by factors' weight. The number of chosen individuals is N/2, where N is the size population (*Irfianti et al., 2016*).

### Crossover operator

Crossover operators are used for generating a new population after the selection operator has chosen half of the population. This operator selects two individuals at random and combines their characteristics to produce offspring for the new population. The uniform crossover method determines whether each of the offspring's characteristics is inherited from one or both parents (*Varun Kumar & Panneerselvam, 2017*).

### Mutation operators

The crossover operator can produce offspring that are strikingly similar to their parents. This problem is solved by the mutation operator, which changes the value of some features in the offspring at random. To determine whether a feature has been mutated, we generate a random number between 0 and 1 (*Deep & Mebrahtu, 2011*).

## The proposed hybrid feature selection (DFGA)

To obtain the best subset of features, we propose a hybrid feature selection technique that utilizes document frequency and genetic algorithms. The proposed DFGA algorithm utilizes the benefits of filter and wrapper feature selection methods. The most relevant attributes are chosen first, based on document frequency. The best subset of features is then selected using a genetic algorithm to obtain the best possible feature subset for text classification. A high-level description of the proposed feature selection method is presented as shown in Fig. 3 below.

## EXPERIMENT

In this section, we investigate the effect of the proposed feature selection on Amharic news document classification. The performance of the proposed feature selection method is

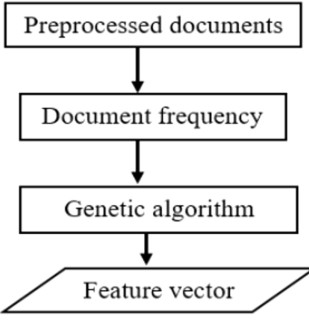

**Figure 3 A pictorial description of the proposed feature selection.**

**Table 2 Genetic algorithm parameters used for this study.**

| Parameters | Value |
| --- | --- |
| Generation | 5 |
| Population | 100 |
| Verbosity | 2 |
| Other parameters | Default |

compared with state-of-the-art feature selection methods in terms of classification. All experiments are run in a Windows 10 environment on a machine with a Core i7 processor and 32 GB of RAM. In addition, the description of parameters used in the genetic algorithm is depicted as shown in Table 2 below.

## Dataset

There is no publicly available dataset for Amharic text classification. Business, education, sport, technology, diplomatic relations, military force, politics, health, agriculture, justice, accidents, tourism, and environmental protection are among the 13 major categories of news used in this study. Each document file is saved as a separate file name within the corresponding category's directory, implying that all documents in the dataset are single-labeled. The news is labeled by linguistic experts of Jimma University. Every document is given a single label based on its content. The dataset consists of documents with varying lengths. The upper bound length of a document is 300 tokens and the lower bound is 30 tokens. So the length of documents in each category is in the range of 30–300 tokens. For each category, we used a number from 1 to 13 to represent the category label. The news categories and amount of news items used in this study are listed in Table 3.

## Performance measure

We assess the performance of classifiers with our proposed document frequency plus genetic algorithm-based feature selection in terms of accuracy, precision, recall, and F-measure.

**Table 3 News categories and the number of news documents in each category.**

| News category | No. of news | Category label |
|---|---|---|
| Business | 257 | 1 |
| Education | 269 | 2 |
| Sport | 251 | 3 |
| Technology | 267 | 4 |
| Diplomatic relation | 270 | 5 |
| Military force | 278 | 6 |
| Politics | 244 | 7 |
| Health | 275 | 8 |
| Agriculture | 256 | 9 |
| Justice | 212 | 10 |
| Accidents | 275 | 11 |
| Tourism | 239 | 12 |
| Environmental protection | 265 | 13 |

### Accuracy

This is the most widely used metric for measuring classifier efficiency, and it is calculated as follows (*Hossin & Sulaiman, 2015*):

$$Accuracy = \frac{TP + TN}{TP + TN + FP + FN} * 100\%. \tag{2}$$

### Precision

It is used to determine the correctness of a classifier's result and can be determined as follows:

$$Precision = \frac{TP}{TP + FP} * 100\%. \tag{3}$$

**A recall** is a metric that assesses the accuracy of a classifier's output. It is calculated using the following equation:

$$Recall = \frac{TP}{TP + FN} * 100\%. \tag{4}$$

The harmonic mean of precision and recall is the **F-measure**, which can be calculated as follows:

$$F-measure = \frac{2 * Precision * Recall}{Precision + Recall} * 100\%. \tag{5}$$

where TP denotes True Positive, TN denotes True Negative, FP denotes False Positive, and FN denotes False Negative.

**Table 4 Performance evaluation of Amharic news document classification using the proposed feature selection method.**

|  | Evaluation metrics | | | |
|---|---|---|---|---|
|  | **Accuracy** | **Precision** | **Recall** | **F-measure** |
| Experimental results in percentage | 89.68% | 89.52% | 89.65% | 89.56% |

**Table 5 Comparison of CTC, RFC, and GBC.**

| No. | Machine learning model | Accuracy (%) |
|---|---|---|
| 1 | ETC | 89.68 |
| 2 | RFC | 87.80 |
| 3 | GBC | 87.58 |

## RESULTS

The results of the experiments are discussed in this section. To determine the best train-test split mechanism for our data set, we conduct experiments using train-test split ratios of 70/30, 75/25, 80/20, and 90/10. The experiment is carried out while all other parameters remain constant. Then we got a classification accuracy of 86.57%, 87.53%, 89.68%, and 87.05% respectively. As a result, we used an 80/20 splitting ratio in all of the experiments, which means that 80% of the dataset was used to train the classifier and 20% of the dataset was used to test the trained model. The proposed feature selection method is evaluated on Amharic news classification on 13 major news categories in terms of accuracy, precision, recall, and F-measure and the result is depicted as shown in Table 4 below.

We also evaluate the performance of the proposed (document frequency + genetic algorithm) based feature selection over different classifiers such as ETC, RFC (Random Forest Classifier), and Gradient Boosting Classifier (GBC). According to our experimental results, ETC outperforms the RFC and GBC classifiers as shown in Table 5 below.

According to the results in Table 5, ETC outperforms RFC and GBC by 1.88% and 2.1%, respectively. We also used ETC to compare DFGA-based feature selection strategies to existing filter feature selection and feature extraction methods like DF, CHI, IG, PCA, hybrid of (IG, CHI and DF) (*Endalie & Haile, 2021*), hybrid of (IG, CHI, DF, PCA) (*Endalie & Tegegne, 2021*) and genetic algorithm. The comparisons of the proposed method with the existing methods were performed using our dataset. The results are shown in Table 6.

The document frequency plus genetic algorithm-based feature selection method produced the highest accuracy, according to the results in Table 6. This is because the proposed feature selection algorithm considers classification accuracy when selecting a subset of features. In our experiment, the accuracy of the proposed feature selection algorithm is 5.44% higher than that of the DF, 15.01% higher than that of the CHI, and 7.13% higher than that of the IG. Furthermore, we discovered that DF outperforms IG and

**Table 6 Comparison between the proposed feature selection methods with existing methods.**

| Learning model | Feature selection | Accuracy (%) |
|---|---|---|
| ETC | IG | 82.73 |
| | CHI | 74.85 |
| | DF | 84.42 |
| | Hybrid of (IG, CHI, and DF) *Akhter et al. (2022)* | 85.82 |
| | Genetic algorithm | 87.21 |
| | PCA | 84.56 |
| | Hybrid of (IG, CHI, DF, and PCA) *Hartmann et al. (2019)* | 88.67 |
| | DFGA | 89.68 |

**Table 7 Comparison of feature selection methods in terms of the number of features.**

| Feature selection methods | Number of features |
|---|---|
| Hybrid of IG, CHI, and DF | 405 |
| Hybrid of IG, CHI, DF, and PCA | 194 |
| PCA | 1,226 |
| GA | 230 |
| DF | 393 |
| DFGA | 100 |

CHI on larger datasets. This is because the probability of a given class and term becomes less significant as the dataset size increases (*Blum & Langley, 1997*).

In addition to classification accuracy, we also compared the proposed feature selection method with the existing feature selection methods in terms of the number of features they produced. As the result, the proposed method produced a minimum number of features as compared with the other method considered in this study. A minimum number of features means, saving the computational time and space taken by the classifier algorithm. The number of features produced by the corresponding feature selection methods is depicted as shown in Table 7 below.

The results indicate the joint use of filter and wrapper methods improves classification accuracy. It also helps to reduce the size of the feature matrix without affecting the classification accuracy. This is mainly because (1) relevant terms are first taken by the filter methods, (2) wrapper methods produced the best subset of features by considering the classifier's performance. Generally, the proposed feature selection method provides the best classification accuracy with the smallest number of features as compared with the existing feature selection methods. This helps us to save the computation complexity.

## CONCLUSION

In this study, we present a hybrid feature selection method that consists of document frequency and a genetic algorithm for Amharic text classification. To validate the performance of the new feature selection strategy, several experiments and comparisons were conducted using various classifiers and state-of-the-art feature selection techniques

such as a hybrid of DF, CHI, and IG, hybrid of IG, CHI, DF and PCA, and GA. The result showed that the proposed feature selection technique gives promising results when we combined it with ETC. As a result, a hybrid of document frequency and genetic algorithm-based feature selection method is suitable for use in a variety of applications requiring Amharic document classification, such as automatic document organization, topic extraction, and information retrieval. We aimed to examine additional categories and datasets, and test the proposed feature selection method on other languages in future work.

## ACKNOWLEDGEMENTS

The authors would like to thank the institute for assisting us with various resources, as well as ENA for providing the dataset for our experiments. The authors would like to express their gratitude to Jimma University for their assistance throughout the research process.

### Funding

The authors received no funding for this work.

### Competing Interests

The authors declare that they have no competing interests.

### Author Contributions

- Demeke Endalie conceived and designed the experiments, performed the experiments, analyzed the data, performed the computation work, prepared figures and/or tables, authored or reviewed drafts of the paper, and approved the final draft.
- Getamesay Haile analyzed the data, authored or reviewed drafts of the paper, and approved the final draft.
- Wondmagegn Taye Abebe analyzed the data, authored or reviewed drafts of the paper, and approved the final draft.

### Data Availability

The Amharic news documents under 13 major news categories (business, education, sport, technology, diplomatic relations, military force, politics, health, agriculture, justice, accidents, tourism, and environmental protection) and the news document in each category are at GitHub: https://github.com/demekeendalie/genetic.

### Supplemental Information

Supplemental information for this article can be found online at http://dx.doi.org/10.7717/peerj-cs.961#supplemental-information.

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
