# Peer review of "Feature selection by integrating document frequency with genetic algorithm for Amharic news document classification"

_PeerJ Computer Science, doi:10.7717/peerj-cs.961_

## Round 0.1 · original submission · Major Revisions

The reviewers pointed out some issues to be fixed. The validity of the findings should be dealt with in the revised version. Please provide a detailed response letter with your revision. Thanks.

·

Basic reporting

This work focuses on the application of Feature Selection by Integrating Document Frequency with a Genetic Algorithm for Amharic News Document Classification. A hybrid of document frequency (DF) and genetic algorithm (GA)-based feature selection method is suitable for use in a variety of applications requiring Amharic document classification. Datasets from different domains with different categories are considered.

First, the writing should be improved a lot. A lot of grammar issues, unsound statements,
and incomplete sentences phrases are very common. there are also very very long sentences, please make it readable, write short and concise sentences.

There are some issues to be fixed.
Introduction: " Examine the performance of the proposed feature selection method in
terms of accuracy, precision, recall, and F-measure." it should be not your main contribution.
What was your starting Research Question or the gap you want to fill? Clearly define your research questions.
Introduction: "2007 census [1]" This is a very old reference, do you have a near time reference for this?

On related work: revise it again even you have not cited the latest paper that was done in
Amharic news document classification like this https://doi.org/10.1371/journal.pone.0251902
Data processing -Normalization: Which is "canonical" normalization or standardization?
What was your base to normalize homophone characters into a single representation?
It would be also nice if your approach is compared with homophone normalization and without homophone normalization approaches.

Results and discussion section (table 6):
Is it the different hybrid results are from the previously conducted paper or from your data?
if it is from the previous paper, the data you have used is different, set clearly your experiment results.

Experimental design

- Even the paper has minor, easily fixable, technical flaws that do not impact the validity of the main results, the paper contributes some new ideas and applicable resources.
- Extensive experiments with multiple feature selection techniques.

- Set clearly the gap that you have filled at the end of your experiments.
- Link your results to the research questions that should be provided.

Validity of the findings

The paper contributes some new findings regarding feature selection for Amharic news document classification.
The dataset that has been provided is also valid and highly applicable for the next research.

Additional comments

Go ahead through the suggestions that are mentioned in the above sections.
Indicate your significant contribution very briefly.

Reviewer 2 ·

Basic reporting

No comment

Experimental design

no comment

Validity of the findings

There is no comparative study of results with other authors [10] on the same dataset. The dataset used in this study does not match the dataset referenced [10].

Additional comments

Line 124, It looks like a regular pre-processing step for any other language. How is the pre-processing different in context to Amharic?

Line 144, Gasser's HornMorpho stemmer: what is the accuracy of the stemmer used? How is it evaluated? is it a Rule-Based or Statistical??

Table 2, what basis are the documents classified into major categories? Who did this classification? Can a document not belong to multiple types, e.g. education and health?

Table 2, what is the length of the documents in each category?
Line 248, Any observations on varying the splitting ratio versus the proposed model's performance?

Can the proposed methodology, etc. used be adapted/used for other similar languages? Conclusion of future work may discuss this.

The dataset used here is small in size, and I recommend that the authors test your proposed algorithm on a larger dataset in a future studies.

Annotated reviews are not available for download in order to protect the identity of reviewers who chose to remain anonymous.

---

## Round 0.2 · accepted · Accept

The paper can be accepted. Congratulations.

·

Basic reporting

No comment.

Experimental design

No comment.

Validity of the findings

No comment.

Reviewer 2 ·

Basic reporting

No Comment

Experimental design

No Comment

Validity of the findings

No Comment

Additional comments

Kindly cite the following papers.

https://doi.org/10.4218/etrij.2019-0458

doi: 10.1109/ICDIM.2018.8847044